# AT-Text: Assembling Text Components for Efficient Dense Scene Text Detection

**Haiyan Li** [1,2] and **Hongtao Lu** [1,*]

1   Department of Computer Science and Engineering, Shanghai Jiao Tong University, Shanghai 200240, China; lihaiyan_2016@sjtu.edu.cn
2   School of Computer Science and Technology, Kashi University, Kashi 844000, China
*   Correspondence: htlu@sjtu.edu.cn

**Abstract:** Text detection is a prerequisite for text recognition in scene images. Previous segmentation-based methods for detecting scene text have already achieved a promising performance. However, these kinds of approaches may produce spurious text instances, as they usually confuse the boundary of dense text instances, and then infer word/text line instances relying heavily on meticulous heuristic rules. We propose a novel Assembling Text Components (AT-text) that accurately detects dense text in scene images. The AT-text localizes word/text line instances in a bottom-up mechanism by assembling a parsimonious component set. We employ a segmentation model that encodes multi-scale text features, considerably improving the classification accuracy of text/non-text pixels. The text candidate components are finely classified and selected via discriminate segmentation results. This allows the AT-text to efficiently filter out false-positive candidate components, and then to assemble the remaining text components into different text instances. The AT-text works well on multi-oriented and multi-language text without complex post-processing and character-level annotation. Compared with the existing works, it achieves satisfactory results and a considerable balance between precision and recall without a large margin in ICDAR2013 and MSRA-TD 500 public benchmark datasets.

**Keywords:** scene text detection; segmentation model; Convolutional Neural Network (CNN); bottom-up mechanism

---

## 1. Introduction

Text reading in scene images is an important task driven by a set of real-world applications, such as image retrieval, license plate recognition, multi-language translation, etc. Scene text detection, followed by text recognition, is an essential procedure of visual text reading. However, due to the varieties of text patterns and complex backgrounds, scene text detection remains a challenging problem.

Among the existing methods for scene text detection, there are mainly two prevalent types: detection-based [1–11] and segmentation-based [12,13] algorithms. The former methods, drawing inspiration from general object detection, design types of anchors to generate candidate regions and filter out false-positive regions to produce accurate bounding boxes for text instances. The latter algorithms, approaching text detection from a segmentation perspective, first adopt the fully convolutional network to perform semantic segmentation and then seek text localizations in segmentation maps. Essentially, to correctly detect the text in the scene image, the localization and boundary of each text instance should be accurately predicted. Detection-based methods usually work well in most cases. However, when there are large-scale variations and multi-oriented text, the anchor designing [1,4,9] and anchor matching [10,11] might be complex and difficult. The semantic segmentation-based algorithms explore a different way and exhibit stronger adaptability to scale the variation and arbitrary shapes of text [12,13]. However, it is difficult to successfully separate each text instance from the dense arrangement in the

segmentation map (Figure 1b). In this case, using post-processing, such as Non-Maximum Suppression (NMS)_, it would difficult to produce accurate boundaries for these text instances. The root cause of the above problem might be that the distinguishing operation of text instances relies on the binarization of the segmentation map. The vague segmentation result and sole binarization threshold might result in such embarrassments.

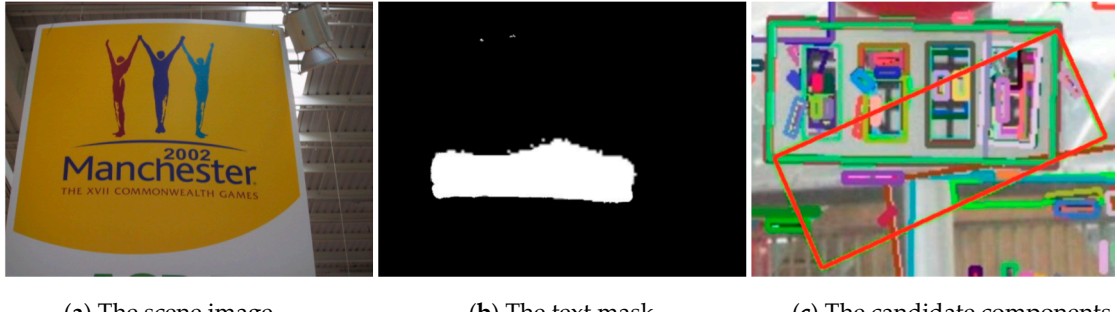

(**a**) The scene image          (**b**) The text mask          (**c**) The candidate components

**Figure 1.** An example of dense scene text detection. (**a**) The input image, (**b**) the text mask obtained by a segmentation model, which produces spurious text instances. (**c**) The candidate components in the bottom-up mechanism—e.g., these components generated by the Maximally Stable Extremal Regions (MSERs) algorithm. There is a component covering part of the text regions and surrounding background simultaneously (shown in red box), which it has been difficult to deal with in post-processing in the previous works.

In this paper, we propose a text detection framework to solve the above problem called AT-text (Assembling Text Components for dense scene text detection). AT-text improves the discrimination of the segmentation map and explores the bottom-up mechanism to alleviate the problem of spurious text instances. To cope with the interference of complex background, we combine the high-level semantic features and low-level appearance features of the segmentation model, achieving a more robust and better segmentation performance. Regarding the challenging of the implicit boundary of dense text instances, we decompose text regions into components for exploring local detail features. In these bottom-up methods, such as in Figure 1c, some generated components are difficult to deal with in post-processing. Therefore, they are generally built on whole-character component detection. We presume each component generally represents a small part of a word/text line, which may include part of a character, a single character, or multiple characters, etc. The segmentation result, such as the text mask in Figure 1b, allows the AT-text to efficiently filter out false-positive candidate components, making it powerful for assembling the remaining text components into different text instances. The contributions in this paper are summarized as follows:

(1)   We propose a novel dense text detection framework in the bottom-up mechanism, without character-level annotation, to generate parsimonious text components and assemble them into word/text lines. This endows the framework with the flexible ability to make use of local information and multi-language text detection.

(2)   We employ a segmentation model that encodes multi-scale text features, considerably improving the classification accuracy of text candidate components. This allows the AT-text to efficiently filter out false-positive components, making it powerful in dealing with densely arranged text instances.

(3)   The experiments demonstrate that AT-text achieves a highly competitive performance in popular datasets and exhibits stronger adaptability to dense text instances and challenging scenarios.

The rest of the paper is organized as follows. Section 2 reviews the related work in scene text detection. The proposed approach is depicted in Section 3. Section 4 presents our experiment and the results of the public datasets. Conclusions and future work are discussed in Section 5.

## 2. Related Works

In the past few decades, scene text detection has attracted a large amount of attention and numerous works have been reported [1–26]. These works have been dominated by bottom-up methods, which are generally built on character component detection. They can be roughly divided into two main groups: sliding window-based methods [1,4,12,13] and component-based methods [2,3,5–8,14,15,17,19,20,24–26]. Sliding window-based methods usually apply some specially designed features to exhaustively search the location of text in raw images [12]. For the sake of the diversity of text patterns, sliding windows or raw images usually adopt a multi-scale strategy to match the text, leading to a large computational burden. Subsequently, manual rules or well-trained classifiers were employed to ensure that the remaining sliding windows contain text. This is good for narrowing down the number of sliding windows. However, manually designed rules inherently limit the capability of handling multi-language and complex backgrounds efficiently. Wang et al. [23] ran a $32 \times 32$ pixel-window over the input images to obtain candidate text lines and then estimated the text locations in the candidate lines. Zhang et al. [1] extracted the symmetry features and appearance features of text with a template consisting of four rectangles to detect symmetry axis as text line candidates. Jaderberg et al. [4] computed a text saliency map based on a sliding window, followed by a character/background classifier. Traditionally, component-based methods commonly contain four stages: component extraction, component analysis, candidate linking, and chain analysis [2]. The first stage usually generates more candidate components by clustering algorithms. The candidate linking stage assembles the character candidates into a pair based on similar geometric and color properties, and then aggregates the pairs into chains. The two analysis stages both employ trained classifiers to verify the components and chains sent from the former stages. Their main difficulties lie in utilizing the discriminative features to train the high-capability classifiers. Therefore, the methods in this group make an effort to extract the discriminative features of text regions.

SWT (Stroke Width Transform) [24] and MSERs (Maximally Stable Extremal Regions) [27] are two representative component-based methods. They extract characters with stroke width or intensity contrast properties. The SWT-based method [2] heavily depends on a single threshold, stroke width, which limits the performance of the detection in varieties of text patterns. Subramanian et al. [25] extracted the strokes of characters based on intensity, spatial, and color-space constraints. Dinh et al. [26] traversed along horizontal scan lines and found that horizontal text lines had a similar vertical stroke width. Boris et al. [24] densely voted on each stroke pixel to obtain a stable identification of strokes and grouped candidates without a brittle process in the other works. MSERs-based methods [5–7,14,17,18,20] progressively calculate intensity contrast by increasing the threshold to control the boundary of connected regions, and hence it is convenient for generating connected components. Due to great improvement of the classification task with Convolutional Neural Network (CNN) [28,29], the deep convolutional neural network is exploited to distinguish text components with a high discriminant ability and strong robustness against a cluttered background [8–13].

To generate more components for high recall, most MSERs-based methods usually apply two strategies. One of these is to preprocess the source images so that the algorithms can function in different variations of data—e.g., multi-channel or multi-resolution data. The other is that the parameters were changed to control the performance of the algorithm [8]. The first strategy aims to increase the number of text component candidates to achieve a higher recall. Ma et al. [7] extracted the components in RGB and LAB color space. Then, a random forest classifier was applied to minimize the false text components. Turki et al. [17] presented a multi-channel MSERs method on enhanced images by edge detection. Their method employed the Otsu and Sobel operations to filter out background pixels and then generated components in several enhanced channels. Finally, two classifiers were used to verify the detection results. Cho et al. [18] applied a variant of the MSERs method to generate components, which were identified by an AdaBoost classifier with two thresholds. Tian et al. [6] produced multi-channel and multi-resolution images for the MSERs method. MSER-generated components were filtered out by a well-trained classifier, followed by a bootstrap

approach to handle the hard-negative samples. The second strategy aims to facilitate the extraction of low-quality texts in the scene image. However, a large number of non-text components and overlapping components are brought about by loose constraints. Therefore, classifiers and heavy post-processing are essential for distinguishing text components. A hierarchical MSERs tree was proposed in Yin et al. [14] to reduce the overlapping components. The components remained after the MSERs pruning was regarded as candidate characters. These candidate characters were computed again by distance weight and a clustering algorithm. The final results were obtained by applying a trained character classifier. In a later work [5], they focused on the oriented text detection algorithm extended from [14]. MSERs-generated components were processed with morphology clustering, orientation clustering, and projection clustering algorithms. Then, two classifiers were trained to estimate the probabilities of characters.

Despite the fact that loose constraint generally improved the recall of low-quality text, it also introduced more non-text components. Therefore, a powerful classifier or filter is required in order to verify the candidate components for high detection precision and decreasing the computation of post-processing. Yao et al. [2] devised a two-layer filtering mechanism to identify the candidate components. The first layer was a filter designed with a set of heuristic rules for statistical and geometric properties. Since some non-text components were still hard to remove by this filter, a classifier was trained with the characteristic scale and major orientation features. Yin et al. [14] trained a character classifier to estimate the posterior probabilities by applying Bayes' rule of text candidates. Jaderberg et al. [4] used Random Forest classifier acting on HOG (Histograms of Oriented Gradients) features [30] to reduce the false-positive word proposals.

CNN has been used in some challenging tasks and obtains significant improvements. It is capable of extracting meaningful high-level features and semantic representations for scene text. Therefore, the classifier based on CNN was adopted to robustly identify the text components in recent works. Huang et al. [8] designed a CNN classifier with two convolutional layers to predict character components generated by MSERs method. The input character candidates were of the fixed size $32 \times 32$. The CNN model showed a high discriminative ability to classify text-like components and greatly improve the performance of text detection. Tian et al. [6] employed a CNN classifier to refine text components. Turki et al. [17] used a Support Vector Machine (SVM) classifier and a CNN classifier to verify MSERs-generated components in multi-channel. Since the binary label was not sufficient to train a robust classifier, He et al. [12] trained a CNN model with multi-level supervised information to teach the model more intuitive character knowledge.

In this paper, the proposed AT-text uses fully convolutional network and the MSERs algorithm that closely related to the aforementioned second strategy. Moreover, we experimented the effect of MSERs constraints in different granularities. The detail of the network design is displayed in Section 3.

## 3. The Proposed Method

In this section, we first show the overall of the AT-text. Next, we introduce the detail of text segmentation network and describe how it can effectively distinguish multi-scale text. Further, the means of generating text components and filtering out algorithm are presented. At last, we depict the implementation of assembling remain components into word/text lines.

### 3.1. Pipeline

The overview of our pipeline is illustrated in Figure 2b. This pipeline is mainly implemented by three parts: a segmentation model for exploiting discriminative text semantics, a text components model for detecting parts of words/text lines, and an assembly model for generating bounding boxes of text instances.

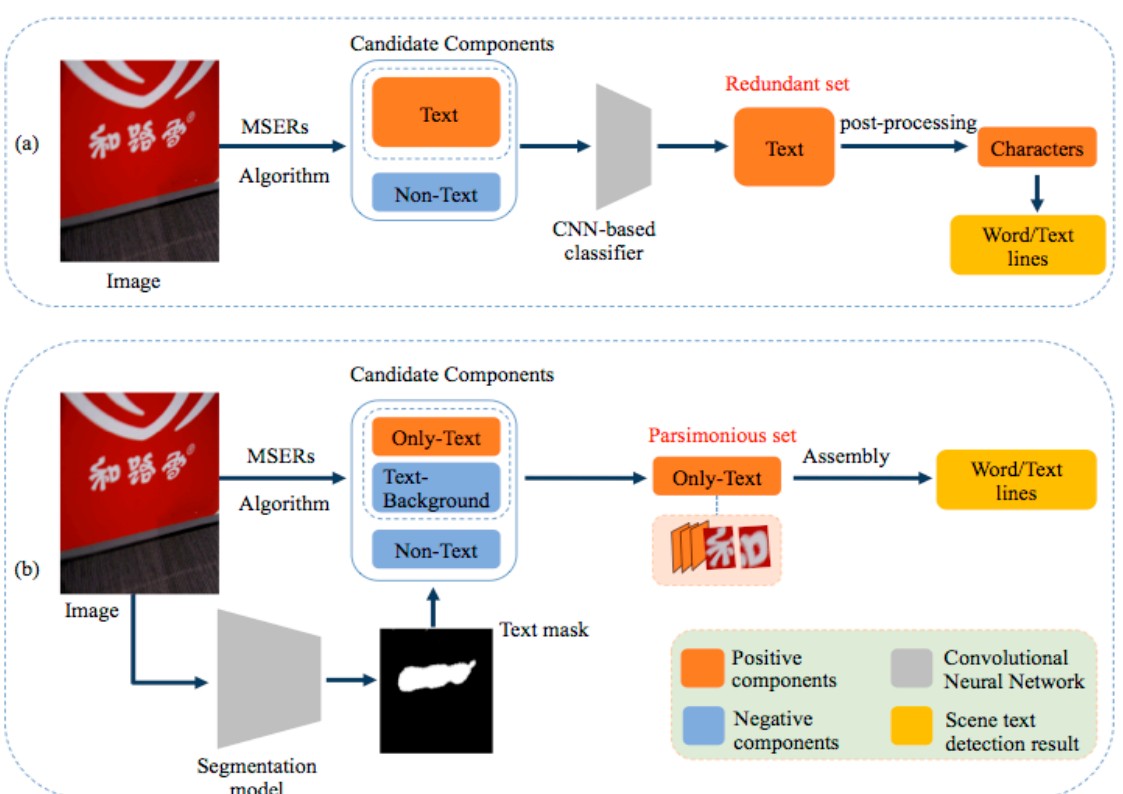

**Figure 2.** Comparison of the pipelines of the previous works and the proposed approach. The top row (**a**) is a common workflow of previous component-based methods. The bottom row (**b**) is the proposed framework.

Figure 2a displays a common workflow of some component-based methods. Based on the assumption that strokes of characters are approximate stable regions with a uniform intensity feature, MSERs detector is used to generate connected components, which are subsequently processed by a classifier (e.g., a shallow CNN classifier) to filter out non-text components. However, the remaining components are still redundant for the construction of words/text lines. As shown in Yin et al. [14], the true text lines accounting for the final text lines is only 9%. Thus, it requires complicated and costly post-processing to remove false-positive components—e.g., obtaining whole character components [5,8,12,14]. We observed that the false positive components are those containing part of the text and part of the background, which are difficult to accurately predict with a classifier.

To deal with the trouble of removing above redundant components, AT-text carefully selects the candidate components to directly supply a parsimonious text component set for the assembly of words/text lines. The candidate components are divided into three categories: Only-Text, Text-Background, and Non-Text, as shown in Figure 2b. Only-Text means that the components are enclosed completely in the text regions. Text-Background components cover part of the text regions and part of the background. Non-Text components are enclosed completely in background. The Only-Text is treated as positive sample, while Text-Background and Non-Text are treated as negative samples to be discarded directly. To this end, we employ a segmentation model to generate text regions mask for selecting the candidate components. Different from previous works, our approach takes advantage of the deep segmentation model to filter out false positive components without multiple steps and complicated manual rules.

### 3.2. Segmentation Model

The text encodes semantic information, which makes it different from background. In this paper, inspired by recently deep learning development, we adopt a segmentation model that particularly

focuses on scene text to provide semantic mask of text regions. To realize this capability, the model is implemented with a fully convolutional network [31] architecture for pixel-wise text prediction.

The details of the architecture are schematically illustrated in Figure 3. The VGG network is employed as a backbone to extract text features. Since the sizes of text regions vary tremendously, determining the existence of large texts would require features from the late-stage of the VGG network, while predicting accurate small text regions needing low-level information in early stages. The convolutional layers, followed by RELU (Rectified Linear Unit), are regularly divided into five stages by four MaxPooling layers. These convolutional layers with $3 \times 3$ convolutional kernel have 64, 128, 256, 512, and 512 kernel numbers, respectively (denoted as $n$ in Figure 3).

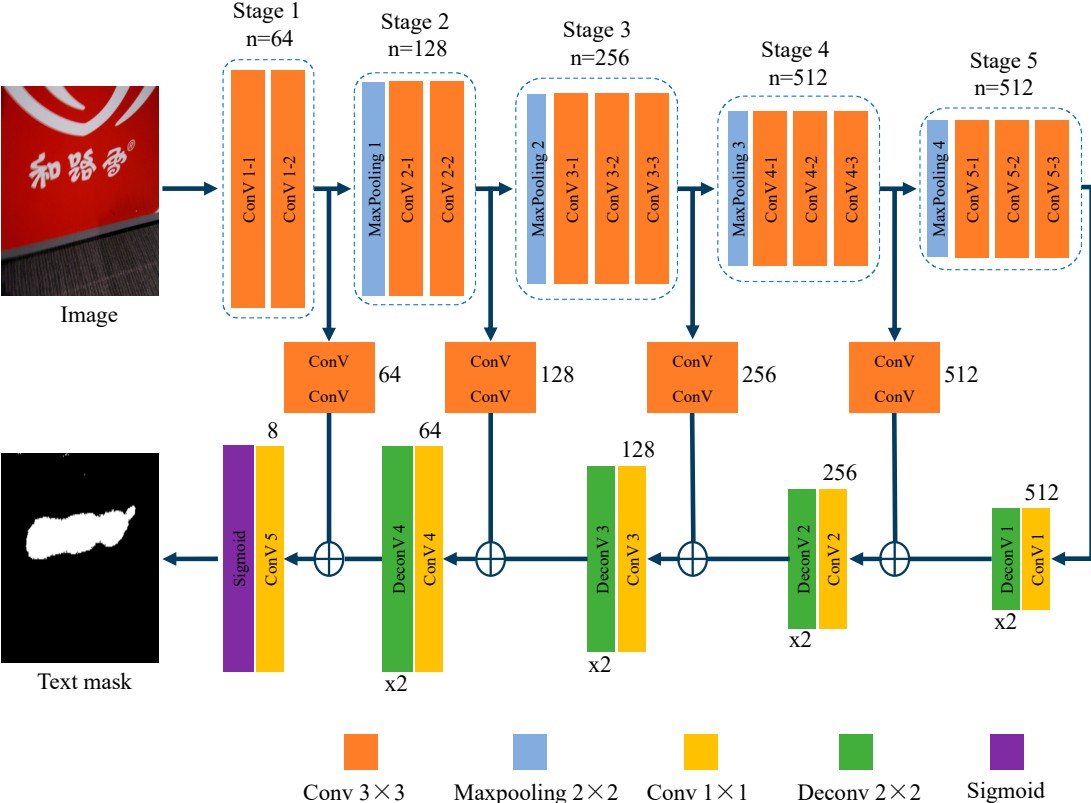

**Figure 3.** The architecture of our segmentation model. The final fusing feature map is in the same resolution as the input image and is converted to the text mask. We use $n$ represents the number of convolutional kernels. The blue layers are Max-pooling, the orange are convolutional layer with $3 \times 3$ kernel size, the yellow are convolutional layer with $1 \times 1$ kernel size, the green is deconvolution layer, and the purple is sigmoid operation.

Different stages have the variable receptive fields and can capture meaningful multi-scale features. The skip connections with two convolutional layers are used to correspondingly combine coarse features and fine features from different stages. Particularly, we select the feature maps before pooling layers to reduce detail information loss. Then, we use $1 \times 1$ convolutional layers and deconvolution layers to up-sample previous coarse outputs. Finally, the $1 \times 1$ convolutional layer and the sigmoid layer are exploited to efficiently generate a pixel-level prediction. Each convolutional layer is followed by ReLU (Rectified Linear Unit) activation. More details of the network can be found in Simonyan et al. [32]. The outputs of the segmentation model are discretized into a binary mask, where values above 0.5 are set to 1 and values below 0.5 are set to 0.

We utilize the training set supplied by the corresponding datasets to finetune the pretrained model. Treated as the binary classification of text/non-text pixel, our method applies cross-entropy

loss formulated as Equation (1), where y text denotes pixel-level text label and p text represents the probability of the text pixel.

$$Loss = -(y\_text \times \log(p\_text) + (1 - y\_text) \times \log(1 - p\_text)). \tag{1}$$

Figure 4 illustrates qualitative results of the produced text masks. It shows that the segmentation model can localize multi-oriented texts, the curved texts in cluttered background, and then provide holistic supervision for identifying Only-text components.

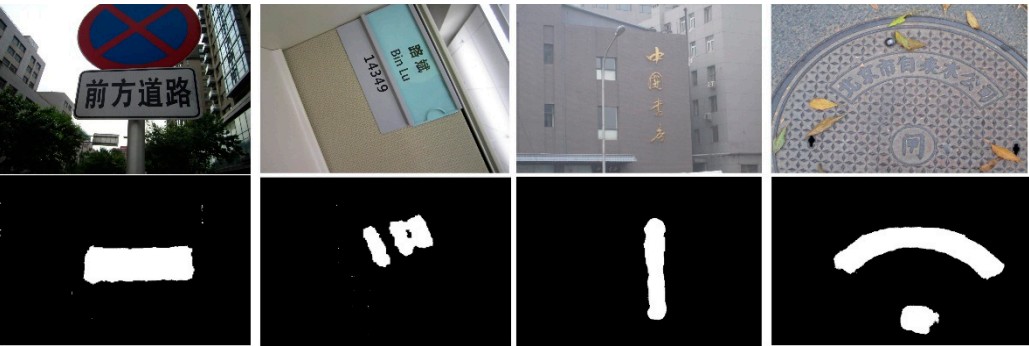

**Figure 4.** The text masks of scene images with cluttered background and multi-oriented texts.

### 3.3. Text Components Generation

An example is shown in Figure 4, where the dense text instances have touching effects in the segmentation mask. Therefore, directly producing the text lines may be difficult or unreliable. We look for a bottom-up manner that is able to extract part of texts and distinguish boundaries using local characteristics of text instances. The MSERs algorithm extracts connected components that remain stable over a range of intensity thresholds. It has a high capability to generate components of various texts, so it has been widely adopted in bottom-up text detectors. However, the methods based on the MSERs algorithm are confronted with two problems. First, as long as a connected component is missed, it would be hard to recollect in the subsequent steps. Therefore, MSERs-based methods prefer to generate more candidate components with weak constraints. Second, the weak constraints might lead to producing more false text components—i.e., covering the texts and the large-area background regions simultaneously (named Text-Background components in this paper). We gather the statistics of MSERs-generated components in Figure 1, as shown in Table 1. Delta ($\Delta$) is the threshold measuring intensity contrast of the stable regions, which equals to 1, 5, 10 respectively. "Total-Number" represents the total number of generated connected components. "Text-Background" is the number of components that contain text regions and background simultaneously. The "Non-Text" can be obtained by calculating the numbers of these three columns. Additionally, we compute the ratio account for "Total-Number" for the clarity of comparisons.

**Table 1.** The statistics of MSERs-generated components with different $\Delta$ on a scene image of ICDAR2013 (International Conference on Document Analysis and Recognition).

| Delta ($\Delta$) | Total-Number | Text-Background | Ratio of Text-Background | Only-Text | Ratio of Only-Text |
|---|---|---|---|---|---|
| $\Delta = 1$ | 31,494 | 6469 | 20.54% | 458 | 1.45% |
| $\Delta = 5$ | 23,702 | 4912 | 20.72% | 311 | 1.32% |
| $\Delta = 10$ | 13,299 | 3138 | 23.60% | 233 | 1.75% |

In Table 1, the total number is huge with either three deltas. Maximum is 31,494 when delta equals 1. Besides this, a smaller delta is able to obtain more components than a bigger one. Shown by Only-Text columns, there are tiny number of components are enclosed by the text regions. The ratio

intuitively indicates this conclusion. When delta equals to 5, only 1.32% of components could be utilized to construct text lines. The large number of Non-Text and Text-Background brings great pressure to the classifier and post-processing in the previous works. Huang et al. [8] selected an average of 516 components to the CNN classifier in one image on ICDAR2011 dataset. In this paper, we further classify the Text-background, almost 20% components, as negative samples to be discarded directly. Moreover, for compensating the less Only-Text, we combine text components generated by the different values of delta in the phase of text line construction.

Figure 5 shows samples of our positive text components. Particularly, the components of Chinese characters are not constrained with the entire character. Therefore, a character with un-connected strokes is extracted by several components to suit the multi-language text detection. Additionally, the proposed approach can deal with different scale text conveniently. Compared with the previous methods, we only provide parsimonious text components for subsequent steps.

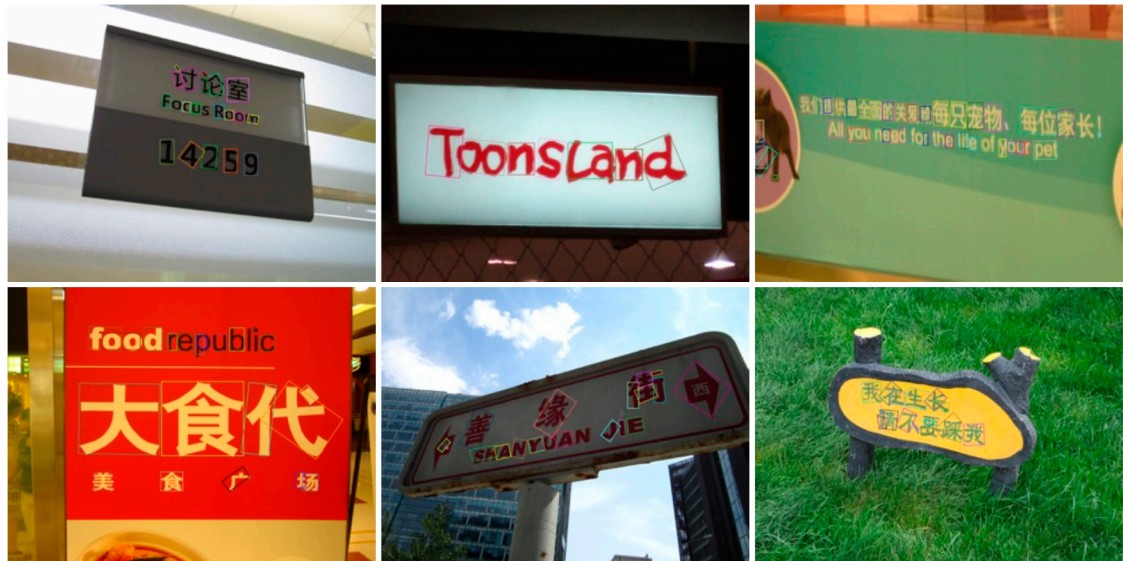

**Figure 5.** The parsimonious text component set distinguished by the meaningful supervision of the text masks.

### 3.4. Assembling Text Components

Benefitting from the parsimonious text component set, we specially design a straightforward approach for assembling positive text components into words/text lines, outputting a single enclosing rectangle for each word/text line. The function minAreaRect in OpenCV could be applied for this purpose. The following steps are applied to construct the final text lines.

Firstly, we sort the text components according to their center points along the scanning direction. In ICDAR2013 dataset, the scanning direction is horizontal. Then, we take a currently un-assembled bounding box as the initial seed and form an initial text line TL. Third, TL iteratively expands by other un-assembled bounding boxes that are similar to the boxes in TL. This process is repeated until this TL no longer changes. Next, we select a new un-assembled bounding box as the new seed to deal with it in the above manner. Finally, when all the bounding boxes are assembled into their text lines, we inspect every pair of text lines; if there exist similar bounding boxes in different text lines, we merge these two text lines into a new one. The merging process repeats until no text line changes. Practically, we further slightly refine the side of the final text line in the horizontal direction according to the contour of the text mask.

Note that one bounding box is considered similar to another bounding box only if (1) the aspect ratio of the two boxes is less than T and greater than 1/T; (2) the minimum horizontal distance between the two bounding boxes should be less than the average distance in the text line; (3) for orientation,

the line connecting two bounding boxes should not deviate from the horizontal direction by an angle θ. In this paper, T is equal to 3. θ is equal to 5 in the ICDAR2013 datasets and 90 in the MSRA-TD500 datasets.

## 4. Experiments

### 4.1. Datasets and Evaluation

We evaluated the proposed approach on two widely used scene text datasets—ICDAR2013 for horizontal text detection and MSRA-TD500 for multi-oriented text detection.

The ICDAR2013 dataset is released for ICDAR (International Conference on Document Analysis and Recognition) competitions [33,34], containing 562 color images with a variety of text patterns, and well captured from the outdoor scene. The texts are horizontal and focused on images center. There are 229 images prepared for the training stage, while 233 images are used for the testing stage. All the images provide word-level annotations. Competition organizers adopt the evaluation algorithm introduced by Wolf et al. [35].

The MSRA-TD500 dataset includes 500 color images, captured from indoor and outdoor scenes with line-level annotation. The texts in images are multi-orientation; with various scales; and consist of two language characters, Chinese and English. Thus, the dataset is very challenging for text detection. There are 300 images used for training and 200 images for testing. The dataset and the evaluation protocol are introduced by Yao et al. [2].

Both two datasets are evaluated by Precision, Recall, and F-measure metrics. The final results are derived from the mean of all the images. Specifically, F-measure, representing the balance of precision and recall, is considered the most important metric of text detection.

### 4.2. The Network Training

The AT-text network aims to predict label of each pixel, but the benchmark datasets supply the coordinates of the rectangle of text regions. To ease implementation, we hypothesize that the pixels inside the rectangles are taken as positive, while the other pixels are negative, meaning background. Although some stroke-surrounding background pixels inevitably gain text label, either our deep convolutional neural network fusing multiple level features or components focusing on connected stroke partly reduce the influence of background noise.

Experiments were implemented in the Keras2.0-based Tensorflow, Opencv2, and Python. The segmentation model was trained on GTX 1080, and 4.10G CPU with 8G memory, optimized by Adadelta. The backbone is finetuned from the ImageNet pre-trained model with data augmentation. The learning rate is set to 1e-3 for all 20K iterations.

### 4.3. The Ablation Study

To verify the effectiveness of the proposed method, we compare the performance of different strategies on the ICDAR2013 dataset, as shown in Table 2. The strategies are summarized as follows:

- Single: exploits the feature map of the last stage of the VGG16 architecture to yield a segmentation mask.
- Fuse: hierarchically fuses feature maps with the same size of the input image, as described in Section 3.3 to encode robust text representation.
- Δ(delta): number in the bracket is the variation in intensity contrast in the MSERs algorithm. Multiple numbers represent that we group all text components generated by these variations.

**Table 2.** Performances of various networks and different delta (Δ) on the ICDAR2013 dataset (%).

| Strategies | Precision | Recall | F-Measure |
|---|---|---|---|
| Single + Δ(10) | 64.03 | 63.36 | 63.70 |
| Single + Δ(1) | 74.89 | 78.04 | 76.44 |
| Single + Δ(1,5,10) | 75.06 | 78.32 | 76.66 |
| Fuse + Δ(5) | 82.24 | 75.48 | 78.72 |
| Fuse + Δ(1) | 86.19 | 80.24 | 83.10 |
| Fuse + Δ(1,5,10) | 87.93 | 80.56 | 84.07 |

Compared with most existing works, AT-text achieves a considerable balance between the precision and the recall without a large margin. From Table 2, we can see the small delta is helpful to improve the accuracy. When the delta changes from 10 to 1, the improvements reach more than 10% in all terms of precision, recall, and f-measure. Grouping all text components with three deltas has a bit of promotion in accuracy than the sole parameter. Additionally, our results demonstrate that it is not necessary to generate more candidate components for a high recall metric in text detection. The strategy of fusing feature maps further improves the performance of baseline architecture greater than 10% in terms of precision. Thus, it is clearly evident that the text masks encoding coarse-to-fine features handle different scale texts gracefully.

### 4.4. Comparison Results

The comparison results of the proposed approach with existing methods on the ICDAR2013 dataset are reported in Table 3.

**Table 3.** Comparison of different text detection methods' performance on the ICDAR2013 dataset (%).

| Methods | Precision | Recall | F-Measure |
|---|---|---|---|
| The proposed | 87.93 | 80.56 | 84.07 |
| Zheng et al. [16] | 89.50 | 77.63 | 83.14 |
| Tian et al. [36] | 85.15 | 75.89 | 80.25 |
| Zhang et al. [1] | 88.00 | 74.00 | 80.00 |
| Zhu et al. [37] | 85.80 | 74.34 | 79.66 |

We achieve the precision, recall, and F-measure values of 87.93%, 80.56%, and 84.07% on the ICDAR2013 dataset, respectively, which shows less margin than most other MSERs-based works [1,16,36,37]. The advancement is significant in terms of recall, which is about 7% better than work with the same precision in Zhang et al. [1]. In Table 3, the other works had a large margin between the precision and recall metrics. Particularly, the recall is less than the precision of about 10%, resulting in a low F-measure value. Text detection with a low recall probably brings extra difficulties in the subsequent text recognition procedure. Figure 6 displays several text masks and text detection results on the ICDAR2013 dataset. Despite the existence of blur, low contrast, and text-liking disturbance, the text masks efficiently identify text locations.

Table 4 shows the performance of the AT-text on the MSRA-TD500 dataset. AT-text achieves a significant improvement in terms of recall compared with other methods. There is also an obvious margin between the precision and recall in the existing works [5,12,13,25,38]. This phenomenon might be explained by the inherent limitation of heuristic rules for detecting a variety of text patterns and oriented text. AT-text gains improvement from two factors. One is that the accurate text segmentation mask supplies rich meaningful supervision to robustly identify Only-Text components, and then avoids designing heuristic rules for filtering out false-positive components. The other factor is that the MSERs algorithm could efficiently detect connected components with detailed information in the relatively simple background, which assists text line construction in a multi-oriented and multi-language task. Some typical results are demonstrated in Figure 7.

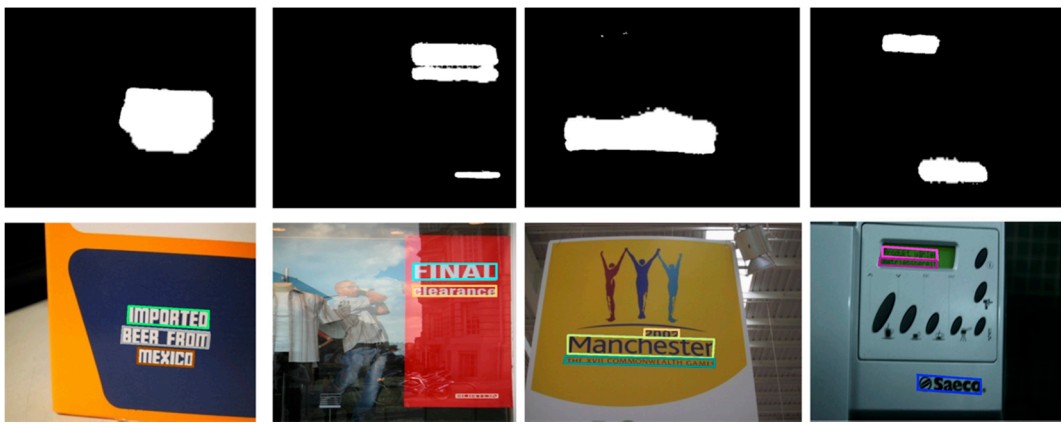

**Figure 6.** The performance of AT-text on the ICDAR2013 dataset. Images in the top row are text masks produced by the proposed segmentation model. Bounding boxes in the bottom row are our text detections.

**Table 4.** Comparison of different text detection methods' performance on the MSRA-TD500 dataset.

| Methods | Precision | Recall | F-Measure |
|---|---|---|---|
| The proposed | 0.77 | 0.69 | 0.73 |
| Yang et al. [38] | 0.95 | 0.58 | 0.72 |
| He et al. [13] | 0.79 | 0.65 | 0.71 |
| Yin et al. [5] | 0.81 | 0.63 | 0.71 |
| Zhao et al. [39] | 0.76 | 0.63 | 0.70 |
| He et al. [12] | 0.76 | 0.61 | 0.69 |

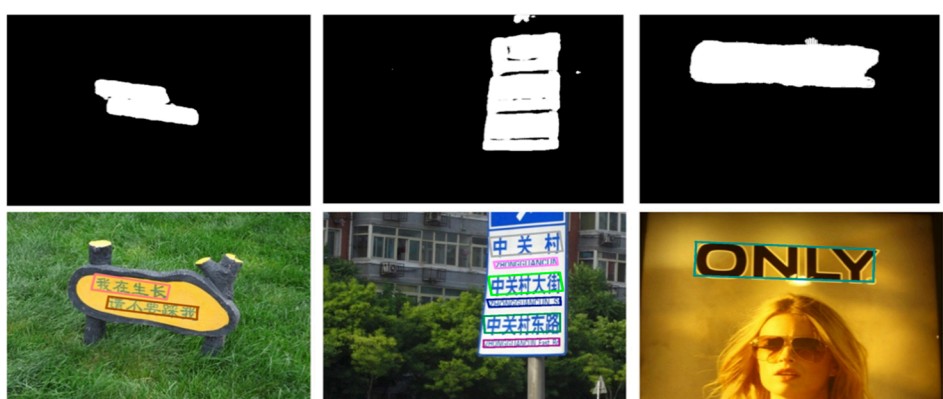

**Figure 7.** The performance of our approach on MSRA-TD500 dataset.

Figure 7 displays several results of produced text masks and text lines grouped with parsimonious components in MSTD-500 datasets, which demonstrate that the proposed method is highly robust to various text patterns and multi-language texts. Our method suppresses the cluttered background and focuses on text regions to supply guidance for correctly predicting multi-scale and multi-language word/text lines.

Figure 8 shows produced the text masks, MSERs-generated components, and final text lines in the three challenging images. The top raw is an example of multi-oriented text detection. The middle raw is a result of the low-quality image with the disturbance of text-like outliers. The bottom is a text line with large character spacing. In the three challenging cases, the text masks supply accurate text region locations to correctly identify positive text components, benefiting from the proposed segmentation model encoding powerful semantic features for multi-oriented text, low-quality text,

and large character spacing text. The remaining components successfully form final text lines without complicated post-processing.

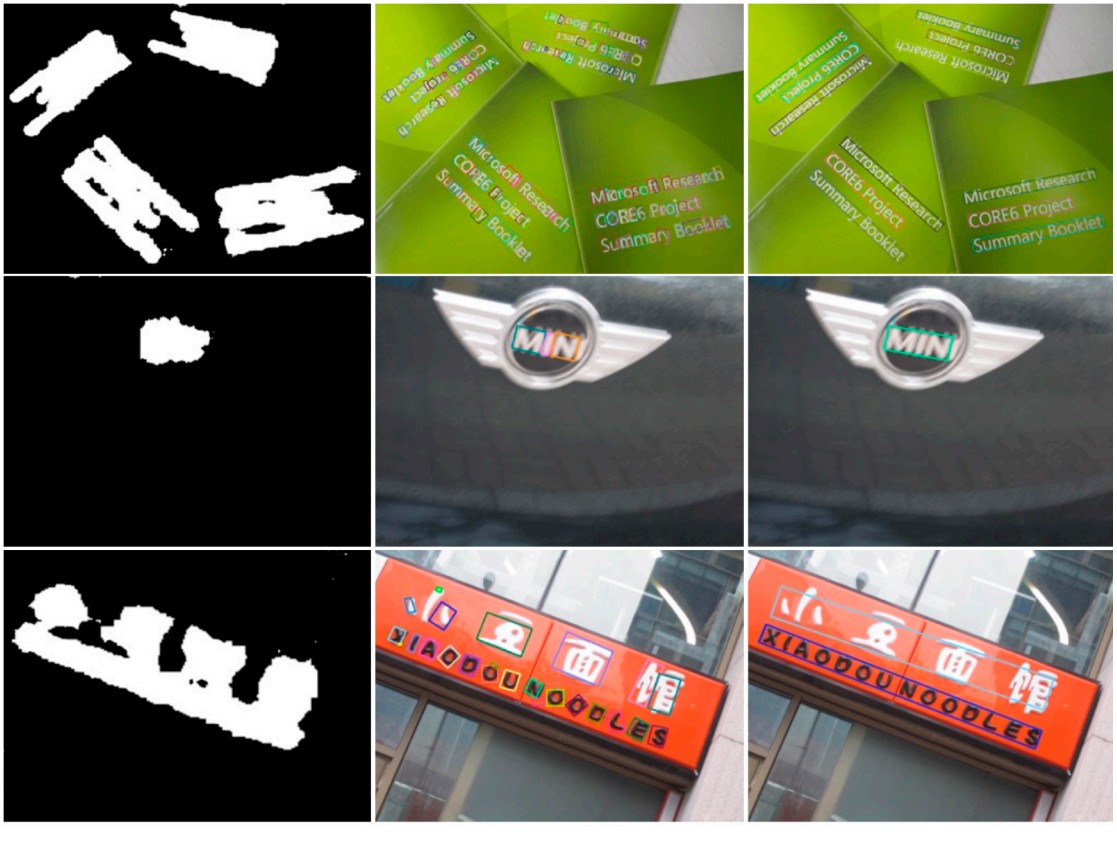

    (**a**) The text masks        (**b**) The positive components       (**c**) The results of text lines

**Figure 8.** The results of text detection on some challenging images. (**a**) Dense text masks. (**b**) Positive text components are successfully distinguished with the text masks. (**c**) The final text lines are constructed directly with our assembling algorithm.

## 5. Conclusions

In this paper, we have presented AT-text, an effective framework for dense scene text detection. AT-text can overcome the problems of previous methods and work well under multi-scale, complex backgrounds; the dense arrangement of text; and other challenging scenarios. It focuses on producing a discriminative segmentation map and the bottom-up mechanism of detecting text components. Without character-level annotation, AT-text can infer the precise bounding boxes of dense text instances. AT-text can effectively alleviate the deficiency of character-level annotation and show a stronger adaptability in detecting multi-language text.

**Author Contributions:** The authors contributed equally to the preparation of the manuscript and the concept of the research. The writing of the draft was carried out by H.L. (Haiyan Li); the review and editing of the draft were carried out by H.L. (Haiyan Li) and H.L. (Hongtao Lu). All authors have read and agreed to the published version of the manuscript.

**Funding:** This research was funded by the Scientific Research Program of the Higher Education Institution of XinJiang. (No. XJEDU2017S043).

**Acknowledgments:** The authors thank the hard work of the editor and the reviewers' comments for improving the paper.

**Conflicts of Interest:** The authors declare no conflict of interest.

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
