# Peer review of "AT-Text: Assembling Text Components for Efficient Dense Scene Text Detection"

_futureinternet, doi:10.3390/fi12110200_

Round 1
Reviewer 1 Report
The authors propose a framework for dense scene text detection.
Some comments:
- Language. The English could be improved.
- Introduction. The introduction is well structured and contains relevant information.
- References. References are adequately discussed in a proper context, where key references on the field are cited.
- Structure. The manuscript is well structured, containing all relevant sections required in a technical paper.
- Results and discussion. Results are appropriately discussed, compared, and presented in the appropriate context to findings against different denoising algorithms.
- Novelty/contribution. I appreciate the hard work done by the authors, and the manuscript presents fair novelty and contributions to the field. Having said that, the following comments bears significance concerning this paper:
- a) Complex backgrounds, variations of text layout and fonts, and the existence of uneven illumination, low resolution and multilingual content present a much greater challenge in the text identification. Your methodology seems to work good in illuminated and high-resolution contents. But how does it work under those conditions?
- b) It was not clear how can text be localized and separated from non-textual information?
- c) What is the time processing? How is this time compared against the other compared methodologies?
- d) What is the next step in your very interesting research?
Author Response
Dear Ms. Li and Reviewer:
Thank you for your letter and for the reviewer’s comments concerning our manuscript entitled “AT-text: Assembling Text Components for Efficient dense Scene Text Detection” (ID: futureinternet-998880). Those comments are all valuable and very helpful for revising and improving our paper, as well as the important guiding significance to our researches. We have studied comments carefully and have made corrections which we hope meet with approval. Revised portions are marked in "Track Changes" in the paper. The main corrections in the paper and the responses to the reviewer’s comments are as follows:
1. Response to comment: (a) Complex backgrounds, variations of text layout and fonts, and the existence of uneven illumination, low resolution, and multilingual content present a much greater challenge in the text identification. Your methodology seems to work good in illuminated and high-resolution contents. But how does it work under those conditions?
Response: Thank you for your comments on our work! Complex backgrounds, uneven illumination, and low resolution are indeed the challenges for the Maximally Stable Extremal Regions ( MSERs) algorithm. The MSERs algorithm can handle text layout and fonts well via extracting the connected components of characters. Hence, we employ Convolutional Neural Network (CNN) to handle complex background, uneven illumination, and low resolution for solving the problems of MSERs. In Figure 5 and Figure 6, we show the results of two scene images in such challenges. It proves our method works well under the above conditions.
2. Response to comment: (b) It was not clear how can text be localized and separated from non-textual information?
Response: In our paper, the segmentation model uses Convolutional Neural Network (CNN) to predict each pixel in scene images and determines whether it is a text pixel. The CNN possesses this ability, it needs to learn a lot of images with annotation and long time training.
3.Response to comment: (c) What is the time processing? How is this time compared against the other compared methodologies?
Response: Our time processing is mainly in the MSERs algorithm. This algorithm calculates the pixel features in repeated regions and needs complex post-processing to select text. The proposed method uses the prediction of CNN to filter out most false positive components of MSERs, as the statistics shown in Table 1. Thus, compared to other comparison methodologies, our method is effective and efficient. Because the time processing is rarely reported in these papers, it is difficult to quantify and compare.
4.Response to comment: (d) What is the next step in your very interesting research?
Response: As explained above, the MSERs algorithm is the computational bottleneck. In our next research, we will exploit other text proposal solutions where proposal computation is cost-free given the whole network's computations.
Reviewer 2 Report
The article deals with an important issue in the area of ​​image detection. The article clearly defines the aims and subject of research. The workflow and the criteria for comparing the proposed method with the solutions known from the literature are clearly described. In my opinion, the paper presents a very good analysis of the obtained results. I have just a few minor comments in the edit area (which will probably be removed at the stage of editing the article before publishing), e.g. there are double spaces in the work, e.g. line 63 between the words semantic features.
Authors should consider comparing the proposed solution with newer publications in this area, e.g. https://arxiv.org/abs/2008.08523, https://ieeexplore.ieee.org/document/8403317, https://www.vlrlab.net/ papers / xu / icg.pdf. I realize that in some cases, a direct comparison will be difficult, but it is worth mentioning at least the broad spectrum of approaches to this topic in the introduction. However, I leave the final decision in this regard to the authors.
Author Response
Dear Ms. Li and Reviewer:
Thank you for your letter and for the reviewer’s comments concerning our manuscript entitled “AT-text: Assembling Text Components for Efficient dense Scene Text Detection” (ID: futureinternet-998880). Those comments are all valuable and very helpful for revising and improving our paper, as well as the important guiding significance to our researches. We have studied comments carefully and have made corrections which we hope meet with approval. Revised portions are marked in "Track Changes" in the paper. The main corrections in the paper and the responses to the reviewer’s comments are as follows:
1. Response to comment: The article deals with an important issue in the area of ​​image detection. The article clearly defines the aims and subject of research. The workflow and the criteria for comparing the proposed method with the solutions known from the literature are clearly described. In my opinion, the paper presents a very good analysis of the obtained results. I have just a few minor comments in the edit area (which will probably be removed at the stage of editing the article before publishing), e.g. there are double spaces in the work, e.g. line 63 between the words semantic features.
Response: Thank you for your affirmation of our work! we delete the redundant space and check the format of other parts of the paper.
2. Response to comment: Authors should consider comparing the proposed solution with newer publications in this area, e.g. https://arxiv.org/abs/2008.08523, https://ieeexplore.ieee.org/document/8403317, https://www.vlrlab.net/papers/xu/icg.pdf. I realize that in some cases, a direct comparison will be difficult, but it is worth mentioning at least the broad spectrum of approaches to this topic in the introduction. However, I leave the final decision in this regard to the authors.
Response: we have read these papers carefully and add them to our Introduction ( The cite numbers are 9,10,11). They are very valuable for improving our analysis. Thank you very much! We modified the content of line 45 as follows: "However, when there are large-scale variation and multi-oriented text, the anchor designing [1, 4, 9] and anchor matching [10, 11] might be complex and difficult. "
PS: Therefore, we change all the cite numbers in the paper.